# Practitioners' views on shared decision-making implementation: A qualitative study

**Anshu Ankolekar**[1], **Karina Dahl Steffensen**[2,3,4], **Karina Olling**[2], **Andre Dekker**[1], **Leonard Wee**[1], **Cheryl Roumen**[1], **Hajar Hasannejadasl**[1], **Rianne Fijten**[1] *

1 Department of Radiation Oncology (MAASTRO), GROW School for Oncology, Maastricht University Medical Centre+, Maastricht, The Netherlands, 2 Center for Shared Decision Making, Lillebaelt Hospital– University Hospital of Southern Denmark, Vejle, Denmark, 3 Institute of Regional Health Research, University of Southern Denmark, Odense, Denmark, 4 Department of Oncology, Lillebaelt Hospital– University Hospital of Southern Denmark, Vejle, Denmark

* rianne.fijten@maastro.nl

## Abstract

### Introduction

Shared decision-making (SDM) refers to the collaboration between patients and their healthcare providers to make clinical decisions based on evidence and patient preferences, often supported by patient decision aids (PDAs). This study explored practitioner experiences of SDM in a context where SDM has been successfully implemented. Specifically, we focused on practitioners' perceptions of SDM as a paradigm, factors influencing implementation success, and outcomes.

### Methods

We used a qualitative approach to examine the experiences and perceptions of 10 Danish practitioners at a cancer hospital experienced in SDM implementation. A semi-structured interview format was used and interviews were audio-recorded and transcribed. Data was analyzed through thematic analysis.

### Results

Prior to SDM implementation, participants had a range of attitudes from skeptical to receptive. Those with more direct long-term contact with patients (such as nurses) were more positive about the need for SDM. We identified four main factors that influenced SDM implementation success: raising awareness of SDM behaviors among clinicians through concrete measurements, supporting the formation of new habits through reinforcement mechanisms, increasing the flexibility of PDA delivery, and strong leadership. According to our participants, these factors were instrumental in overcoming initial skepticism and solidifying new SDM behaviors. Improvements to the clinical process were reported. Sustaining and transferring the knowledge gained to other contexts will require adapting measurement tools.

**Data Availability Statement:** Data cannot be shared publicly as participants did not give consent for their transcripts to be shared in a public repository. Data are available from MAASTRO

Clinic's Clinical Trial Office (contact via info@maastro.nl) for researchers who meet the criteria for access to confidential data.

**Funding:** This study was supported by a Technology Transfer Grant (TTG) from the European Society for Radiotherapy and Oncology (ESTRO - www.estro.org). AA was the recipient of this mobility grant. The funders had no role in study design, data collection and analysis, decision to publish, or preparation of the manuscript.

**Competing interests:** The authors have declared that no competing interests exist.

## Conclusions

Applying SDM in clinical practice represents a major shift in mindset for clinicians. Designing SDM initiatives with an understanding of the underlying behavioral mechanisms may increase the probability of successful and sustained implementation.

## Introduction

Active participation by patients in the clinical decision-making process is central to patient-centered care and can be achieved through shared decision-making (SDM) [1]. SDM has the following characteristics: (1) the clinician and patient are both involved in the decision-making process, (2) both parties exchange information with each other, (3) treatment preferences are exchanged, and (4) the final treatment decision is agreed upon by both clinician and patient [2].

These elements have been formalized by Elwyn et al. into what has come to be known as the traditional "three talk model" framework [3]. This model states that patient autonomy is built on providing high-quality information and supporting the deliberation process by processing the information and arriving at an informed treatment decision. This is achieved through three steps: the choice talk, the option talk, and the decision talk. In the choice talk, the clinician makes the patient aware that there is a choice to be made between several options (in a revised version of this model this talk is referred to as a team talk to emphasize the goal of inviting the patient as a partner in the decision-making process [4].) Next, the option talk consists of the clinician providing information on these various (treatment) options. Finally, in the decision talk, the options are examined in light of the patient's preferences and a decision is made together when the patient is ready.

Recently, methods such as the SHARE approach have been developed to make the SDM process more intuitive for practitioners [5]. The SHARE approach converts SDM into a five-step process: Seek the patient's participation, Help the patient understand their options, Assess their values and preferences, Reach a decision together, and Evaluate the decision. Although the elements of the SHARE approach correspond largely with existing models such as the three-talk model, using a five-step approach is beneficial for training purposes as it assists clinicians in remembering and applying the core elements of SDM [6].

Yet, while these models seek to simplify the SDM process and despite positive intentions, half of clinicians do not put SDM training into practice [7]. SDM implementation is impeded by numerous barriers, both on the clinician side and the patient side. The biggest barriers cited by clinicians are (in order of frequency): time pressure, lack of applicability due to patient characteristics, lack of applicability due to the clinical situation, a perception that patients do not want to participate in decision-making, and unwillingness to ask patients about their preferred level of participation [8]. Patient-reported barriers center on logistical factors (such as a perception that clinicians have busy schedules and a perceived lack of continuity between different clinicians/departments) and consultation factors (certain patient characteristics, trust, a perceived power imbalance [9].)

Despite these barriers, a few institutes have been successful in implementing SDM in clinical practice. In Denmark, the Center for Shared Decision Making at Vejle Hospital in the region of Southern Denmark (www.cffb.dk) was one of the forerunners of a national initiative to bring policymakers, health care professionals, and patient organizations together to make healthcare more patient-centered [10]. With a focus on improving patient-centeredness in

oncology, the Center successfully implemented and evaluated SDM, supported by patient decision aids (PDAs) in paper format, in the care process for several cancer patient groups, to the extent that it is now being used as a model for wider regional implementation. Evaluations of the Center's SDM initiative have shown an increase in patient engagement with no significant increase in consultation length in SDM groups compared to usual care in lung cancer and breast cancer, as well as significant reductions in decisional conflict and decisional regret for patients undergoing lung cancer diagnostics [11–13]. In addition, in a Danish national survey of patient experiences, patients rated Vejle Hospital's oncology center above the national average on most fronts, particularly clinician efforts to involve patients in decisions [14]. Till date, the perspectives of the clinicians that practice at Vejle hospital have not yet been examined and this remains a gap in the Center's existing evaluation of the program [15]. Understanding which aspects of the SDM implementation process clinicians find challenging and which strategies help overcome these challenges can add to our overall understanding of how to design and execute implementation initiatives that motivate clinicians to adopt these new ways of working.

The purpose of this study is to gain insight into practitioners' experiences of the introduction of a SDM initiative into their workflow at Vejle Hospital, where 'practitioners' in this context refers to individuals involved in the SDM initiative either as a clinician, nurse, team leader, or researcher. Our study was guided by three research questions centered on: (i) practitioners' initial attitudes towards SDM as a concept; (ii) their experience of the implementation process and factors that influenced implementation; and (iii) perceived outcomes. Such insights add to the current understanding of the motivational mechanisms that affect SDM implementation and may help inform similar implementation efforts in other contexts and regions.

## Methods

### Study design

We conducted a qualitative study consisting of semi-structured interviews with practitioners working at Vejle Hospital. The interview guide was created based on prior findings in the literature on barriers and facilitators to SDM and PDAs as well as prior research at MAASTRO Clinic that explored this theme [8]. In this previous MAASTRO study, oncologists had participated in semi-structured interviews to determine the barriers and facilitators for SDM and a PDA, both of which were yet to be implemented. For the present study, this interview guide was modified to account for the fact that Vejle Hospital had implemented SDM and PDAs, but the focus on barriers and facilitators remained. The modified interview guide (provided in S1 Appendix) consisted of questions along the following themes: (1) the participant's experiences with usual care and initial impressions of the SDM paradigm, (2) experience of SDM training, (3) using paper PDAs in practice, (4) challenges in putting SDM into practice, (5) effects of SDM on the consultation process and outcomes, (6) implementation success factors and remaining challenges.

In keeping with the semi-structured nature of the data collection, minor modifications to the interview guide were also made following the first 2–3 interviews. This was to allow participants to expand more on aspects of the implementation that they found relevant.

At the time of the study, KDS was practicing as an oncologist and researcher. KO, AD, LW, CR, and RF were researchers and AA and HH were PhD students. AA, KDS, KO, CR, and RF had prior experience in conducting qualitative research. AA, AD, LW, KDS, and RF conceived the study design and scope.

## Study population

Participants were identified through KDS who was leading the Center and recruited by means of purposive sampling [16]. The main criterion was that the participant should be well-acquainted with SDM, either through direct clinical practice or general awareness by virtue of having a strategic/leadership role. In order to obtain a range of perspectives, both oncologists and oncology nurses were approached (Table 1). Ten participants agreed to be interviewed; five oncologists from four specializations, and five participants with a nursing background. The sample is reflective of the Center's strategy of applying SDM in a wide range of disease areas (aside from what are typically considered preference-sensitive conditions) and for smaller decisions along the treatment trajectory, such as nutritional decisions made with nurses. Seven participants (four oncologists and three nurses) had undergone the Center's SDM training and were applying SDM with paper PDAs in routine practice. Aside from their clinical experience, three of the participants played leadership roles in SDM implementation; two leaders were an oncologist and nurse pair based within the Center and responsible for implementation and evaluation efforts.

## Ethics

According to the National Danish Consolidation Act on Research Ethics Review of Health Research Projects, Consolidation Act number 1083 of 15 September 2017 section 14 (2), notification of questionnaire surveys and medical database research projects to the system of research ethics committee system is only required if the project involves human biological material. Thus, the study was conducted without an approval from The Committee on Health Research Ethics, as approval for this kind of study is not required according to Danish law [17].

In accordance with general research ethics, potential participants were sent an electronic invitation with information about the purpose, aims, procedure, and the voluntary nature of the study. This information was also conveyed verbally at the start of the interviews. Participants were informed that the interview would be audio-recorded for transcription purposes and that they would be sent a written transcript of their interview with the opportunity to review it and make clarifications if desired. Participants' consent to take part was obtained

**Table 1. Participant characteristics.**

| Characteristic | No. of participants (N = 10) |
|---|---|
| **Gender** | |
| Female | 8 |
| Male | 2 |
| **Experience (years)** | |
| 6–10 | 2 |
| 11–15 | 6 |
| 16–20 | 2 |
| **Professional background** | |
| Oncology | 5 |
| *Lung cancer* | *2* |
| *Breast cancer* | *1* |
| *Gynaecological cancer* | *1* |
| *Colorectal cancer* | *1* |
| Nursing | 5 |

prior to recording and was repeated on tape. Several measures were taken to preserve confidentiality; audio recordings were stored in a secure location and accessible to one researcher (AA) who performed all the transcription. Any personally identifying information was removed from the transcripts and each transcript was assigned a numerical identifying code.

## Data collection

The interviews were conducted by AA who was a visiting researcher throughout the duration of the study and had little exposure to the clinic context beforehand, thus was in a position to analyze its SDM implementation from an external perspective. AA was not acquainted with the study participants prior to the start of the study, apart from initial acquaintance with KDS from whom she obtained approval to visit the clinic and conduct the study. Through KDS and KO, AA first made the acquaintance of potential participants in person and then followed up via email with a brief outline of the study purpose, procedure, information on participation, and an invitation to participate.

In the period between March 2020 and December 2020, seven interviews were conducted on site at the participant's clinic/office and three were conducted via video-conference due to the COVID-19 pandemic. All participants who were approached participated and there were no participants who dropped out of the study. Only the researcher (AA) and the participant were present in the interviews. All interviews were audio-recorded. The average duration of the interviews was 49 minutes (range: 30–71 minutes) and all interviews were transcribed verbatim from the recordings by AA.

After the interviews, participants were sent their transcript via email and given the opportunity to make changes or comments; one participant added supplementary information to their transcript. The coding process was started after the first interview and minor modifications were made to the interview guide, such as adjustments in the order, structure, and formulation of questions. Data collection continued until no new themes emerged from the participants' responses, as per the coding framework that was being constructed in parallel.

Besides the qualitative interviews, KDS arranged site meetings with SDM training developers, PDA designers, project managers, and other researchers to familiarize AA with the clinic setting. In addition, AA observed KDS's patient consultations to gain a better understanding of how SDM and PDAs are applied in practice.

## Data analysis

Data was analyzed using the thematic analysis method under a reflexive approach as described by Braun and Clarke [18]. This method consists of six phases: becoming familiar with the data, generating initial codes, searching for themes, reviewing themes, generating definitions and names for each of them, and writing up the results.

The first phase consisted of reading the transcripts line by line in order to gain familiarity with the data. Next, for each transcript, initial codes were generated for fragments of text. These codes described or summarized the content of the text fragment. AA coded all the transcripts, and HH independently coded three transcripts. AA and HH then systematically compared the codes they had independently generated for each text fragment and discussed their interpretations of the text. If necessary, codes were rephrased or adjusted, and based on these discussions the list of codes generated across the entire dataset was built. This coding was performed manually in Microsoft Word and the coded extracts were transferred to a Microsoft Excel document in preparation for the next phase.

In phase 3, AA and HH grouped the codes into categories according to potential themes and sub-themes. Themes were derived from the data rather than identified in advance. These

themes were reviewed in phase 4 by first re-reading the coded text fragments associated with each theme and checking whether there was coherence between the text fragments within themes, i.e. whether each theme accurately reflected the contents of the underlying text fragments. After establishing coherence within themes, the dataset was re-examined in light of the themes to check that they were representative of the data as a whole. Subsequently in phase 5, themes were refined and discussed among multiple researchers (AA, HH, RF, and CR) to ensure that they were understandable and coherent.

Analysis was conducted based on a collaborative reflexive approach in which Interpretations of the data were based on the researchers' interpretation of the data and prior theoretical knowledge. Our approach was inductive and prioritized engaging with the data and discussing multiple interpretations as opposed to achieving a specific consensus.

While it is not possible to completely eliminate interviewer bias, efforts were made to reduce confirmation bias during analysis by having another researcher (HH) independently perform coding on a selection of interviews. This selection included interviews of clinicians, nurses, and a participant in a leadership position so that a range of perspectives were coded. The potential for leading questions and wording bias was reduced by adapting an interview guide that had been tested on multiple clinicians in a similar study conducted at MAASTRO Clinic.

## Results

The following section presents the themes that emerged under each of the three research questions: practitioner attitudes towards SDM, factors that influence implementation, and perceived outcomes. Our findings revealed that participants' attitudes regarding SDM prior to its implementation at Vejle Hospital ranged from skeptical to receptive. Four themes emerged under factors that influenced SDM implementation: raising awareness, leadership, reinforcement of new SDM behaviors and habits, and flexibility in PDA delivery. Strong leadership was related to the themes of raising awareness and reinforcement. Participants noted the effects of the SDM implementation on their experience of the clinical process and how measurement methods may need to be adjusted to ensure that SDM implementation is sustained. These themes and the related subthemes are summarized in the thematic map in Fig 1 and elaborated upon below.

### Attitudes towards SDM

During the pre-implementation phase, participants recounted that although they and their colleagues were generally positive towards SDM, there was still a degree of resistance.

**Initial skepticism.** Three of the clinicians recalled being initially skeptical about SDM, citing doubts about whether patients would be knowledgeable enough to actively take part in clinical decision-making, and whether the SDM steps might be too restrictive and unnatural for consultations. One participant recalled feeling that SDM might be at odds with clinicians' traditional training of ensuring progression-free survival. Another doubted whether patients would understand the consequences of their decision over the long term, and for this clinician this issue remains a concern till date.

*"I've been working with this for 15 years plus, and I give [patients] 15 minutes to make the decision. Do they actually know what is the difference between side-effects and liver metastasis?" (Participant 1)*

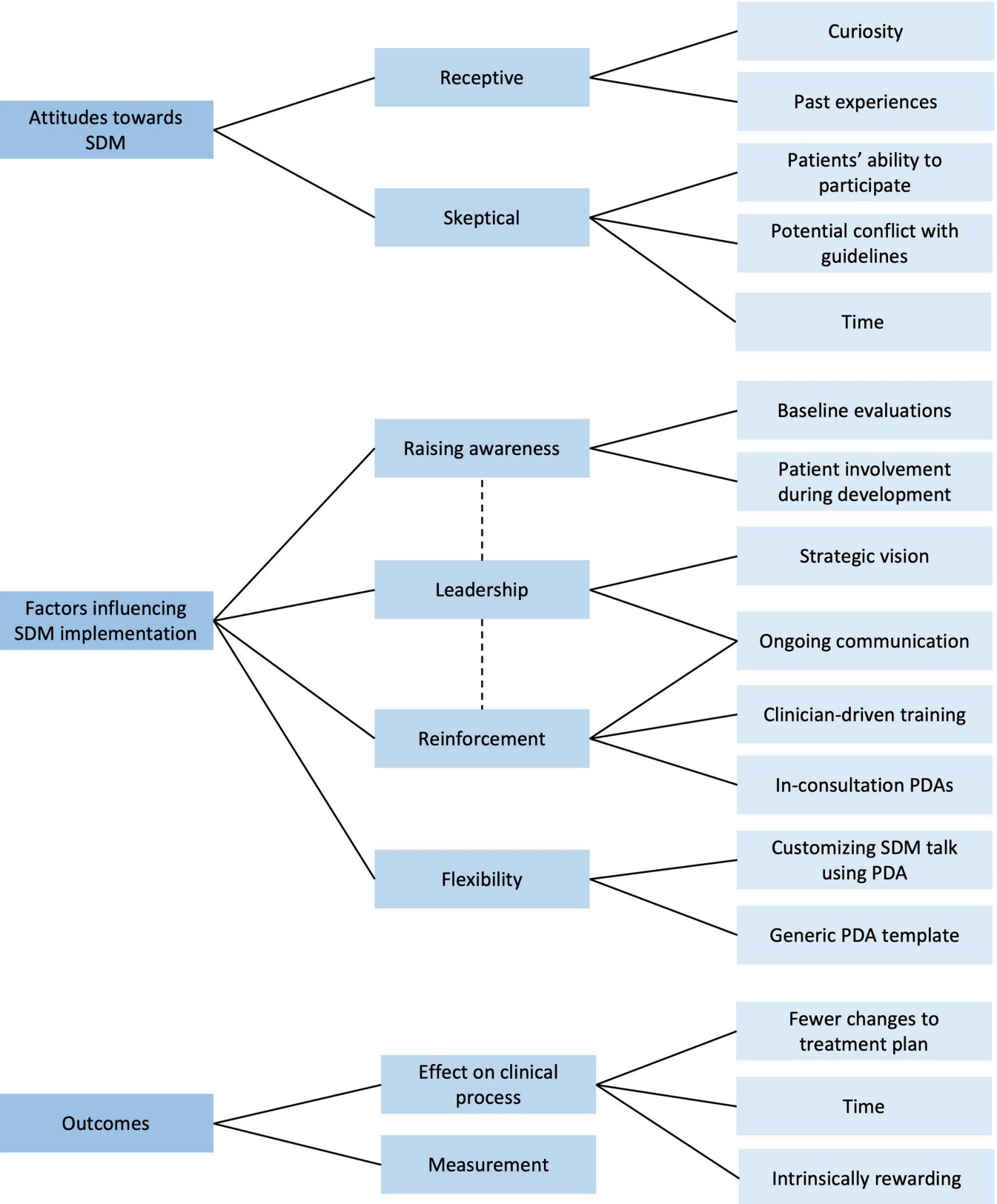

**Fig 1. Thematic map of themes and subthemes for each research question.** Solid lines indicate links between themes and subthemes, while dashed lines indicate relationships between themes.

**Receptive.**   In contrast, four participants, primarily those with a nursing background, said that they had a positive and optimistic feeling about SDM due to their experience in seeing the long-term impact on some patients whose preferences were not incorporated in the decision. One participant described following up with an anal cancer patient who was severely affected by high dose radiotherapy and was unaware that a lower dose with less severe side-effects was an option. This participant described how hearing about SDM years later immediately triggered the thought of that patient and how much their suffering could have been avoided if the clinician had presented the option of a lower dose. For these participants, involving the patient in the decision felt like a natural and logical course of action.

"*I remembered [the patient] said 'If you just had told me that I had two different options, maybe I could have chosen otherwise'. She was so sad about her situation. So, it just made sense to me to work with [SDM].*" (Participant 10)

Aside from the points mentioned by clinicians interviewed, three participants who played a leadership role in the implementation also mentioned typical objections raised by clinicians in general, such as busy schedules, time constraints, and resistance to changing their way of working. They mentioned several actions that were taken to lay the groundwork for SDM implementation. These actions form the basis of the themes that emerged for part (ii) of the research question, namely practitioners' views on the factors that influenced SDM implementation.

## Factors that influenced implementation

**Raising awareness.**   Before developing the SDM training course, the Center systematically measured the current level of patient participation and clinician efforts to involve patients by means of the OPTION-12 instrument [19]. The results were shared with clinicians, not for the purpose of evaluating individual clinicians but to give a general indication of which SDM behaviors were present in consultations and which behaviors may need to be developed. Two participants who were in a leadership role emphasized the importance of letting clinicians see SDM as an enhancement of their current practice rather than an interruption or a completely different way of working. Showing clinicians which aspects of SDM they were already performing set the stage for learning additional SDM techniques formally. For three participants, showing clinicians concrete evidence of current practice was a key step in shifting attitudes.

"*You need to tell [clinicians] the good stories, you know the carrot. I think the carrot works much better than the stick.*" (Participant 9)

Second, involving patients alongside clinicians in PDA development helped clinicians realize that patients experience the decision-making process rather differently from what they assumed. One leader with a nursing background recalled that "including patients in that process also moves something for the clinicians. They suddenly become aware that patients are not thinking the same [way] as [clinicians] are." (Participant 10).

**Leadership.**   Four participants emphasized the importance of strong leadership in successful implementation, beginning with the hospital's strategic vision of putting the patient first in all interactions. This led to the idea that SDM should be implemented across the trajectory, i.e. not only for major treatment decisions but also for smaller decisions made with nurses, such as nutrition. Therefore, the idea of SDM was brought not only to clinicians but to nurses and other support staff as well so that the patient would experience continuity and a consistent experience of being involved at all levels.

*And that has to be communicated from the top. I mean, from the leaders, that 'That is what we use here, and this is what we want to [do]. This is the way we want to talk with our patients.'" (Participant 3)*

One nurse team leader also emphasized the importance of empowering clinicians to take charge of the training process and teach their peers without too much top-down management as "there's something occurring when management enters in a teaching session. . . the power of the room kind of shifts a little bit. I think it's important that you keep it as a teaching room." (Participant 3)

Aside from defining the strategic vision, the involvement of leadership in practical matters was also considered important. These included keeping SDM on the agenda at staff meetings, ensuring that PDAs are being kept up-to-date and new staff members are receiving training. Ongoing communication between leadership and clinicians was considered particularly important. For example, several participants mentioned that it was sometimes a challenge for clinicians to come to a consensus regarding the risk information and statistical evidence the PDAs should contain, even though most patients considered this the least important aspect of the deliberation and decision-making process. In such situations, leaders played a key role in refocusing the discussion to what matters to patients.

**Reinforcement.** Nearly all participants described the biggest challenge in the implementation phase was for clinicians and nurses to learn new habits and adjust their way of working. Moving from the pre-implementation phase to the implementation phase involved conducting ongoing training courses and putting SDM into action in the clinic with the support of PDAs. This was initiated at a smaller scale at first in certain disease areas and therefore not all clinicians across the hospital had received SDM training. Seven of the participants had undergone the Center's SDM training course and described it as a one-day course added on to the mandatory two-day communication skills course for clinicians. The training was conducted by a trained clinician-nurse pair who had undergone a separate two-day SDM course based on the Train-the-trainer model https://www.zotero.org/google-docs/?sr4RG7 [20]. The course covered the theoretical aspects of SDM, role-playing, and interactive sessions to get insights from clinicians on how to teach their peers. This course was based on the SDM model conceptualized by Elwyn et al. and was tested and refined by means of a participatory action research study that involved clinicians, patients, researchers, trainers and other stakeholders [21]. The central idea behind the training was that it should be led by clinicians themselves rather than external educators. Two participants mentioned the benefit of groups of clinicians learning SDM together and from each other as a shared experience that helped reinforce the teachings.

*"It took some effort, but the other doctors and the nurses around me [did] the same course, so we had the same language about it. And in my experience, it was like re-visualizing my conversation. So it's more fun being a doctor this way." (Participant 7)*

The tangible paper PDAs were regarded as a key element in solidifying the knowledge gained in the training course and putting it into practice. One leader recalled that making sure the PDAs would actually be used in the consultation was a key consideration in their design and format. The risk of "over-the-counter" PDAs that are given to patients for use at home is that clinicians may consider it a quick fix and assume that the patient has been informed about their options sufficiently through the PDA. This may reduce the incentive to formally go through the SDM process and make it easier to relapse into traditional decision-making. For this reason, the Center opted to make the PDAs physical in-consultation tools. Four participants described how the PDA served not only as a supporting tool for the SDM process but as

a learning tool for clinicians; having a tangible PDA provided a structure for the SDM conversation and a way to move from one step to another.

*"Sometimes I think that [the PDA] is mostly to help the doctors. . . . And for me, it's also easier in my mind to structure the conversation. And then the clue is to use the [PDA] from the beginning." (Participant 7)*

**Flexibility.**   The paper PDAs were designed according to the traditional three-talk SDM model [3]. Each PDA was in the form of a folder with five components corresponding to the five steps of the SDM model, and contained multiple cards describing the benefits and harms of each treatment option, patient stories, and statistics about survival and side-effects. The cards could then be spread out on the table to provide both patient and clinician with a shared visual overview of the information and treatment options. Some participants reported that this gave them the flexibility to tailor the SDM conversation to the factors that mattered to the patient, for instance by moving the cards around in order of preference or removing cards that the patient felt were not relevant to them.

Three participants mentioned the challenges in translating knowledge about SDM from one context to another, including building new PDAs for different disease areas and keeping them up-to-date. At Vejle Hospital, this challenge was addressed by placing a generic version of the paper PDA on a web-based platform for clinicians and nurses to edit. This gave them the flexibility to update existing PDAs based on new clinical evidence or create new PDAs for different decision-making contexts.

## Outcomes

When participants reflected on the outcomes of the SDM initiative on their practice, two themes emerged: improvements in the clinical process and measurement.

**Effects on the clinical process.**   Participants considered the prime benefit of SDM to be that the consultation was more focused towards what is relevant for patients. Learning about the patient's lifestyle, activities, and future plans, among other things, allowed the clinician to frame information in terms of impact on the patient's daily life and avoid overloading the patient with clinical jargon. One participant described the process of engaging with each patient as an individual intrinsically rewarding, while another felt that sharing the process with the patient makes the clinician's job easier due to the confidence that the right decision is being made for the patient. Two participants observed better treatment compliance and fewer changes to the treatment plan.

*"My [anecdotal] experience is that by performing SDM whenever you have a new patient, the [treatment] plan you make is more often being completed. I used to adapt plans much more often earlier because they didn't tolerate the treatment or they were not confident that it was the right thing. They needed extra consultations." (Participant 7)*

When asked about the effects of SDM on consultation length, participants said that SDM consultations were not significantly longer than usual care in their experience. According to two participants, SDM might take some extra minutes but they emphasized that it is worth the time investment to make a better decision. In addition, choosing the right treatment from the start may mean fewer consultations needed in the future. A third benefit was that patients feel more confident about treatment decisions and exhibit better compliance, lending support to

the idea that a lengthier consultation may not be a barrier to SDM if there is a visible improvement in clinical outcomes.

*"It maybe takes a few minutes more, but then the thing is, isn't that okay? Isn't it okay to spend a few minutes? You're going to take this treatment, maybe chemotherapy for 4–6 months. You're going to have radiotherapy, endocrine treatment for five or 10 years, I think two minutes [more]—that's okay . . . that we spend a little bit of time if we have better compliance, if the patient feels they're more involved." (Participant 1)*

**Measurement.**  Experiencing SDM in practice also led some participants to reflect on changes that may be needed at the organizational level for sustained implementation. One participant noted that current performance metrics are based on activity, such as the number of consultations performed, and do not account for the fact that patient choices may deviate from standard procedure under SDM. This participant recognized the need for value-based rather than activity-based metrics and emphasized the importance of cooperation and communication between department managers/supervisors and clinicians.

Transferring the knowledge gained from local implementation to the regional level and ensuring that implementation stays on track over the long-term was seen as another set of challenges. Current methods to evaluate SDM focus on the quality of the clinical interaction and outcomes. Some participants felt that an additional multidimensional measure of SDM implementation may be needed in order to evaluate the progress at the department level. Possible dimensions mentioned were the level of clinician skills, organizational/cultural readiness, and leadership. The emphasis was on understanding local contexts and adapting the Center's knowledge and experience accordingly.

*"We always have to adapt to the culture. So we can't just, you know, do a complete package and say, 'Here you go, you should do this.' We have to adjust all the time." (Participant 10)*

## Discussion

Although healthcare is becoming increasingly patient-centered, SDM is yet to become the norm in most countries [22]. Even when clinicians are enthusiastic about patient engagement, misperceptions about what constitutes SDM, what distinguishes it from informed decision-making, and a lack of organizational support can thwart implementation efforts [23]. As a result, SDM and PDA uptake may fail to be sustained over time [24]. Within this context, we studied SDM implementation at Vejle Hospital, Denmark, where, under the leadership of the Center for Shared Decision Making, SDM and PDAs have been implemented with considerable success [11–13]. The purpose of this qualitative study was to examine practitioners' experiences of the Center's efforts to introduce SDM and PDAs into clinical practice. In doing so, we provide insights into the factors that may play a role in the hospital's sustained success in implementing SDM. Our analysis revealed challenges to SDM spanning the pre-implementation to post-implementation phase and the strategies that were used to address them. These include overcoming clinician resistance, ensuring that training and tools are successfully put into practice, and facilitating sustained implementation efforts over the long-term through appropriate performance metrics and continuous communication between the various stakeholders (clinicians, nurses, team leaders, and researchers).

Frequently cited objections to SDM include doubt about its applicability to the clinician situation or the individual patient and perceptions that it will disrupt busy workflows [25,26].

These perceptions were present in the pre-implementation phase in our study and can result in resistance towards adopting a different way of conducting consultations [27]. There was a pronounced difference in our results in how clinicians and nurses viewed patient participation; nurses were enthusiastic about SDM due to their first-hand experience seeing the consequences of not taking patients' wishes into account in treatment decisions, whereas for clinicians this shift happened after working with patients on developing the PDAs and was reinforced through repeated application of SDM principles. Previous findings from the Center have shown that pre-implementation OPTION scores indicated lower levels of SDM [11]. Communicating this disparity to clinicians may be the first step in shifting attitudes, as clinicians' perception that SDM improves their practice makes sustained implementation more likely [28].

Interestingly, while time constraints are the most frequently cited barrier to SDM in the literature, this objection was not present in our results. A previous study conducted at the Center which assessed the impact of SDM and PDAs on consultation length found no significant increase, and in fact a decrease in variability of consultation lengths [12]. The authors speculate that the structured nature of the SDM consultations owing to the PDA format makes consultations more standardized in terms of procedure. The results of similar studies using in-consultation PDAs suggest that this format may be ideal for clinicians concerned about time constraints [29,30].

Another factor in increasing clinician motivation was empowering clinicians to take the lead in learning and training each other in SDM, particularly with minimal direct involvement from management. The Center has previously described its approach to creating ownership through involvement and this approach extended to its SDM training course, which is based on the Train-the-trainer model in which clinicians become trained in training their colleagues, thus becoming ambassadors for SDM [15,20]. This approach has been associated with positive results in various contexts, with the importance of a clinical ambassador or champion being emphasized in multiple studies [31–34]. Currently, there is no consensus in the literature regarding which training approach works best for teaching SDM in a clinical setting; measuring training effectiveness is a complex endeavor because there is no generally accepted best measure of SDM [35]. Training evaluation metrics tend to focus on the level of knowledge clinicians gained and indirect outcome measurements such as impact of the training on patient satisfaction and quality of the clinical interaction, while neglecting clinicians' evaluations and satisfaction of their training [36]. While outcome measures are important metrics, it is also necessary to assess clinician satisfaction with the training itself, as positive appraisal has a significant impact on the motivation to learn from the training and put its principles into practice [37].

Moving from training to regular practice also requires measures to facilitate the development of new habits, provision of appropriate resources, and social support [38]. Previous findings revealed that even in spite of positive intentions, only half of clinicians put SDM training into practice [7]. Our findings suggest that the form and function of PDAs played a key role in this regard. A number of our participants considered the PDA to be as much a supporting tool for clinicians as it was for patients. Using a tangible PDA that is structured according to the steps of the SDM model can help reinforce the SDM training through repetition. Evidence from the Center shows that between two groups of clinicians who had received SDM training, the level of patient engagement and SDM measured by OPTION scores was significantly higher when a PDA was used [11]. This suggests that while training on its own may improve clinicians' knowledge and skills to an extent, the repetition of using the five-step PDA may help solidify the contents of the training till it becomes routine [39]. There is preliminary evidence of PDAs being used as 'scaffolding' to facilitate SDM consultations, and certain PDA

formats such as decision cards are specifically designed to function as collaborative tools rather than purely information tools [40,41]. However, we are unable to draw firm conclusions from this result as the overwhelming majority of PDAs are designed for patients to use outside the consultation [42].

Although PDAs are increasingly being offered in a digital format, preliminary evidence suggests that, given a choice, patients tend to use paper PDAs more frequently than digital PDAs and rate them higher in terms of overall satisfaction [43,44]. While both formats increase patient knowledge and users of digital PDAs tend to perform the values clarification function at a higher rate, the limited evidence still points to a digital divide in which there is a preference for print-based PDAs. Practitioners aiming to use PDAs as a supporting tool for SDM would therefore benefit from investigating their patients' needs and preferences regarding decision support format as well as considering what role PDAs would play in the clinical workflow, i.e. whether they are deployed purely as an informational tool outside the consultation or used within the consultation to guide the SDM process.

It is also important to note that tools such as PDAs, while useful as supporting material, cannot compensate for the relational aspect of the SDM process. The core of SDM is effective communication between patients and clinicians [45]. Furthermore, it may not be possible to have a PDA for each clinical situation, and events such as the COVID-19 pandemic, in which some consultations must be conducted remotely, further underscore the need for empathic patient-centered communication [46,47]. This extends beyond the patient-clinician interaction to the way patients interact with the healthcare system as a whole. Steps taken to build patients' trust ensure that SDM initiatives do not operate in isolation at the point of decision-making but become part of a culture of care that prioritizes the patient at all points of the pathway [48]. According to our participants, Vejle hospital's 'Patient First' vision was the impetus for its SDM initiative, suggesting that SDM was a means to an end rather than an end in itself. This distinction has been emphasized by several authors, and empirical evidence highlights the importance of a shared vision, i.e. a specific 'end', in effective implementation [49–51]. This theme has not been extensively explored in the SDM context and could be a valuable area for future research.

The importance of leadership in sustaining implementation efforts over time was emphasized by multiple participants in our study. Leadership that is unsupportive of change can be a barrier to implementation, and leaders play a key role in shaping the environments that favor sustained application of training and PDAs over the long-term [52]. Alongside the regular SDM training for clinicians, the Center also developed a 3-hour training program geared towards leaders to acquaint them with the principles of SDM to help foster a positive implementation environment, according to K. Dahl Steffensen, MD, PhD (written communication, June 2021). Such initiatives can support the creation of a "learning health care system" in which performance metrics are aligned with the organizational vision for SDM implementation [53]. For instance, involving patients through SDM can shift previously 'traditional' treatment patterns based on clinical guidelines to more individualized patterns based on patient preferences; some patients may prefer less intensive treatments, some may choose to forgo certain treatments altogether, or otherwise deviate from guidelines [54]. Health care systems must account for this. Clinical departments that are evaluated using activity-based metrics, such as number of procedures performed or the number of consultations conducted, may have misaligned incentives that hinder SDM implementation [55]. This issue is increasingly on the policy agenda in several countries around the world, including Denmark, although the evidence of current value-based models is limited [56]. While hospital managers and team leaders must be aware of such models and their implications within their own contexts, wider actions at the policy level are needed to put such changes into motion.

## Strengths and limitations

This study was conducted in a context in which SDM has been relatively successfully implemented and it is therefore possible that participants had mostly positive attitudes towards SDM and the type of PDAs used owing to their experiences. As such, our study may be affected by selection bias. We have attempted to get a range of perspectives by interviewing participants with varying degrees of exposure to SDM principles, however the small-scale nature of this study allows us to draw tentative conclusions on how best to support SDM initiatives among clinicians.

Qualitative research is often said to lack generalizability, and another limitation of our findings could be the extent to which they hold up in other contexts and countries. Cultures vary even within different hospitals and departments, and organizational culture is considered to be the most crucial factor that influences implementation success [57]. There is a lack of studies exploring specific strategies to change organizational culture in order to improve health care processes, but the critical role of leadership has been well-established [58,59]. Culture can emerge by default or be built purposefully, and it is important for leaders at all organizational levels (from hospital administrators to team leaders) to understand their role in shaping the culture [60,61]. Furthermore, the success factors we identified (training, tools, and leadership) do not operate in isolation, and so an effective implementation strategy must consider the interplay between the various factors identified [57].

Another limitation of our study is that we focus exclusively on the clinician perspective of SDM implementation and we do not include the patient perspective which is central to SDM. As a result, while our findings give insight into factors that may help clinicians practice SDM, we are not able to comment on how patients experience the process. Further research is needed as patients do not always feel involved or experience SDM despite clinicians' best efforts [62,63]. Although patients were heavily involved during the development and design phase of the Center's SDM initiative, further qualitative research into patients' experiences of the SDM process may provide a more comprehensive picture of successful implementation. Possible research questions may include how involved patients felt in their treatment decision under the SDM process, their experiences with using the paper PDAs, and the impact of their choices on their quality of life.

## Conclusion

This study describes practitioners' insights and experiences regarding SDM implementation in their practice. We found that prior to implementation, many clinicians harbored a degree of skepticism about the value and practicality of SDM. Under factors that facilitated SDM implementation, four main themes emerged: raising awareness, leadership, reinforcement, and flexibility. Systematically measuring the level of patient involvement prior to SDM implementation helped make clinicians aware of the gap between current practice and an ideal SDM process. Supporting clinicians by means of role-play training and in-consultation PDAs helped in reinforcing the new habits associated with SDM so that implementation could be sustained. Sustainability was reinforced by creating an online platform where generic PDA templates could be customized by clinicians and/or nurses, allowing for greater flexibility and adaptability to different clinical contexts. Strong leadership was instrumental in driving these changes.

In conclusion, putting SDM principles into practice successfully requires clinicians to learn new behaviors and actions. This process may be facilitated by combining a top-down and bottom-up approach in which clinician leaders guide SDM initiatives while empowering clinicians to create supporting tools and learn from each other.

## Supporting information

**S1 Appendix.**
(DOCX)

**S1 File.**
(DOCX)

## Acknowledgments

The authors thank the staff at the Center for Shared Decision Making for facilitating this study as well as the participating clinicians and nurses who volunteered their time and insights.

## Author Contributions

**Conceptualization:** Anshu Ankolekar, Karina Dahl Steffensen, Leonard Wee, Rianne Fijten.

**Data curation:** Anshu Ankolekar.

**Formal analysis:** Anshu Ankolekar, Hajar Hasannejadasl.

**Funding acquisition:** Anshu Ankolekar, Karina Dahl Steffensen, Andre Dekker, Leonard Wee.

**Investigation:** Anshu Ankolekar, Karina Dahl Steffensen, Leonard Wee.

**Methodology:** Anshu Ankolekar.

**Project administration:** Karina Dahl Steffensen, Karina Olling, Andre Dekker, Leonard Wee.

**Resources:** Karina Dahl Steffensen, Karina Olling, Leonard Wee.

**Supervision:** Andre Dekker, Leonard Wee, Rianne Fijten.

**Validation:** Karina Dahl Steffensen, Hajar Hasannejadasl.

**Visualization:** Cheryl Roumen.

**Writing – original draft:** Anshu Ankolekar.

**Writing – review & editing:** Anshu Ankolekar, Karina Dahl Steffensen, Karina Olling, Andre Dekker, Leonard Wee, Cheryl Roumen, Rianne Fijten.

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
