## [Decision Letter · Decision Letter 0]

17 Aug 2021

PONE-D-21-20569

Clinician motivation in shared decision-making implementation: A qualitative study

PLOS ONE

Dear Dr. Fijten,

Thank you for submitting your manuscript to PLOS ONE. After careful consideration, we feel that it has merit but does not fully meet PLOS ONE’s publication criteria as it currently stands. Therefore, we invite you to submit a revised version of the manuscript that addresses the points raised during the review process.

We look forward to receiving your revised manuscript.

Kind regards,

Chaisiri Angkurawaranon

Academic Editor

PLOS ONE

2. Please provide the interview guide used.

4. Thank you for stating the following in the Funding Section of your manuscript:

“This study was funded by the ESTRO Technology Transfer Grant (TTG).”.

We note that you have provided funding information that is not currently declared in your Funding Statement. However, funding information should not appear in the Funding section or other areas of your manuscript. We will only publish funding information present in the Funding Statement section of the online submission form.

“This study was supported by a Technology Transfer Grant (TTG) from the European Society for Radiotherapy and Oncology (ESTRO - www.estro.org). AA was the recipient of this mobility grant. The funders had no role in study design, data collection and analysis, decision to publish, or preparation of the manuscript.”.

Reviewers' comments:

Reviewer's Responses to Questions

**Comments to the Author**

1. Is the manuscript technically sound, and do the data support the conclusions?

Reviewer #1: Partly

Reviewer #2: Partly

2. Has the statistical analysis been performed appropriately and rigorously? 

Reviewer #1: N/A

Reviewer #2: N/A

3. Have the authors made all data underlying the findings in their manuscript fully available?

Reviewer #1: No

Reviewer #2: Yes

4. Is the manuscript presented in an intelligible fashion and written in standard English?

Reviewer #1: Yes

Reviewer #2: Yes

5. Review Comments to the Author

Reviewer #1: This is a well-written and interesting manuscript on clinicians’ experiences with implementing SDM within a hospital setting that is known for successful use of SDM in routine practice.

I have two primary concerns with this manuscript. First, the purpose/focus of the study appears to change throughout the manuscript and is thus unclear. At times, it seems that the authors are interested in learning about clinicians’ experience of the implementation process (e.g., descriptive accounts of the SDM training and PDA development process), while at others it appears that the authors are specifically focused on perceived barriers and facilitators to SDM implementation. Further, the title of the manuscript leads one to believe that the focus is on factors that motivate clinicians to implement SDM, which is not consistent with the more comprehensive focus of the manuscript. The authors should further clarify the purpose of the study and ensure that it remains consistent throughout.

Second, themes derived from the data are briefly mentioned in the results section, but not elaborated upon in a coherent way. After mention of the themes, results are organized according to experiences at various stages of the implementation process rather than by theme. Further, Figure 1 does not appear to map on to any of the named themes. This makes evaluation of how themes were derived and what data support them difficult. The authors might consider a re-organization of the results section to better elucidate themes.

Additional comments on specific sections of the manuscript are below.

Abstract:

The abstract should clearly describe all themes that were derived from the data.

Methods:

Citations are needed when referencing specific methods (e.g., thematic analysis).

How was it determined that participants had the relevant experience with SDM implementation in order to participate effectively in the interviews? Was the decision to include both those who had undergone SDM training and those in leadership strategic (i.e., part of the purposive sampling strategy)?

p. 11 (lines 177-181): This paragraph is unclear. Were only open codes compared? Or both open and axial codes? How were open codes further categorized and who performed this step?

Results:

For context, it would be helpful if the SDM training were described in greater detail. For instance, what SDM model was this training based on?

Reviewer #2: Review report

Manuscript Number: PONE-D-21-20569

Manuscript Title: Clinician motivation in shared decision-making implementation: A qualitative study

This manuscript addresses an important theme and is well written. The relevance for this study to the practice field is high. A major revision is needed regarding the analysis and presentation of the results, in addition to some minor revisions. I do believe that this can become a good contribution to the field.

Abstract and introduction

1) Abstract p.2: “The goal was to identify implementation barriers, strategies used to address them, and remaining challenges.” Introduction p.6: ” The aim of this study, therefore, is to gain insight into how clinicians at Vejle Hospital experience the introduction of a SDM initiative into their workflow, the training process, and the impact on their daily practice.” Discussion p.20: “The purpose of this qualitative study was to examine practitioners’ experiences of the Center’s efforts to introduce SDM and PDAs into clinical practice.” The aim/ goal/purpose of the study should be consistent throughout the manuscript.

2) The research question should be reported.

Methods

3) I recommend the authors to give the description of the authors’ contribution under each related section, for example “The interviews were conducted by AA..” could preferable have been described under the heading data collection. “Participants were recruited by means of purposive sampling [16] and were identified through KDS..” is related to study population and could preferable have been described under this heading.

Ethical guidelines:

4) The contribution of the authors AA, KDS and HH is described but the role of the other authors is unknown. What was their contribution in this study? This is important to describe according to the Vancouver recommendations.

5) Anonymization of the participants is important in research and the authors should outline how anonymity is facilitated.

Data collection:

6) The interview guide is presented with two themes numbered 5.

7) Please report who transcribed the interviews.

8) The authors report a study consisting of semi-structured interviews. This design is not compatible with “observations were recorded as field notes” (p.11). If the field notes are included in the data material in addition to the interviews the study design should be revised.

Analysis:

9) Please include a reference to thematic analysis method.

10) It is unclear how the field notes were analysed. Did the author mix the two different data sets, the interviews and the field notes? Can the authors elaborate this?

11) Please elaborate what the interpretations were based on.

12) The last section in “Data analysis” focuses on the interviews and should be moved to related section (data collection).

Results, discussion, conclusions

13) The quote on p.16 is not identified (participant no.)

14) It is unclear if the results reflect the data material (the participants’ answers of the 6 themes in the interview guide). The aim of this study was to gain insight into how clinicians at Vejle Hospital experience the introduction of a SDM initiative into their workflow, the training process, and the impact on their daily practice. Only “The challenges encountered by our participants at each phase and the strategies they used to address them are summarized in Figure 1 and elaborated below” (p.12) are presented as results. The authors are advised to look at the entire data material in light of the research question (which must be presented in the manuscript) and make an analysis were the results reflect the whole data material and the aim of the study.

15) Figure 1 presents an overview of challenges to SDM implementation and strategies to overcome them at different phases of implementation. It is unclear if these items are the themes, categories and codes. The authors are encouraged to provide a description of the coding tree. There should be clarity of themes were the themes are clearly presented in the results and clarity of categories and codes with a description of diverse cases or discussion of these in the results.

16) The conclusion should answer the aim and research question and must be reconsidered.

17) The patient perspective is the onus of SDM. This study reflects only the clinicians’ perspective and should have been reported as a limitation and discussed.

6. PLOS authors have the option to publish the peer review history of their article (what does this mean?). If published, this will include your full peer review and any attached files.

Reviewer #1: No

Reviewer #2: **Yes: **Lise Sæstad Beyene

---

## [Author Response · Author response to Decision Letter 0]

15 Oct 2021

Reviewer #1: 

This is a well-written and interesting manuscript on clinicians’ experiences with implementing SDM within a hospital setting that is known for successful use of SDM in routine practice.

I have two primary concerns with this manuscript. First, the purpose/focus of the study appears to change throughout the manuscript and is thus unclear. At times, it seems that the authors are interested in learning about clinicians’ experience of the implementation process (e.g., descriptive accounts of the SDM training and PDA development process), while at others it appears that the authors are specifically focused on perceived barriers and facilitators to SDM implementation. Further, the title of the manuscript leads one to believe that the focus is on factors that motivate clinicians to implement SDM, which is not consistent with the more comprehensive focus of the manuscript. The authors should further clarify the purpose of the study and ensure that it remains consistent throughout.

Second, themes derived from the data are briefly mentioned in the results section, but not elaborated upon in a coherent way. After mention of the themes, results are organized according to experiences at various stages of the implementation process rather than by theme. Further, Figure 1 does not appear to map on to any of the named themes. This makes evaluation of how themes were derived and what data support them difficult. The authors might consider a re-organization of the results section to better elucidate themes.

Authors’ response:

We thank the reviewer for highlighting the two general issues regarding the study’s focus and the presentation of the results. We have made the following changes:

- The research question has been formulated more explicitly in the final paragraph of the Introduction, namely that the purpose of the study is to investigate practitioners’ views of SDM implementation in their practice, in particular: their initial attitudes towards SDM, factors that supported them in implementing SDM, and perceived outcomes. These changes can be found on pg 5, lines 79-86. The title of the manuscript has also been adjusted to reflect the aim of the study. 

- The results section has been re-organized based on the three components of the research question described above. We have described the themes that emerged under each of the components and replaced Fig 1 with the thematic map that resulted from our analysis (pg 12).

Additional comments on specific sections of the manuscript are below.

Abstract:

The abstract should clearly describe all themes that were derived from the data.

Authors’ response:

In the ‘Results’ subsection of the abstract, we have added the four themes that emerged as factors facilitating the SDM implementation, namely: raising awareness, reinforcing the new behaviors associated with SDM, flexibility in adapting the SDM process to different contexts, and leadership. This information has been added on pg 1,lines 14-22.

Methods:

Citations are needed when referencing specific methods (e.g., thematic analysis).

Authors’ response:

We have added a reference to Braun and Clarke’s thematic analysis method on pg 10 (lines 168-169) and added a more detailed description of the six-step thematic analysis process in the ‘Data analysis’ subsection (pg 11, lines 172-199).

How was it determined that participants had the relevant experience with SDM implementation in order to participate effectively in the interviews? Was the decision to include both those who had undergone SDM training and those in leadership strategic (i.e., part of the purposive sampling strategy)? 

Authors’ response:

The criteria used in the purposive sampling strategy was that the participants should be well-acquainted with SDM, either through direct practice, such as clinicians, or general awareness of the concept, such as those in a more strategic leadership role (pg 7, lines 109-112).

p. 11 (lines 177-181): This paragraph is unclear. Were only open codes compared? Or both open and axial codes? How were open codes further categorized and who performed this step?

Authors’ response:

We have rewritten the paragraph so that each phase of the thematic analysis process is described along with the researchers who performed the steps (pg 11, lines 172-199).

Results:

For context, it would be helpful if the SDM training were described in greater detail. For instance, what SDM model was this training based on?

Author’s response:

We have added further details on pg 17 (lines 301-303) on the SDM training, which was based on the SDM model conceptualized by Elwyn et al (2012) and developed on the basis of a participatory action research study. 

Reviewer #2: Review report

Manuscript Number: PONE-D-21-20569

Manuscript Title: Clinician motivation in shared decision-making implementation: A qualitative study

This manuscript addresses an important theme and is well written. The relevance for this study to the practice field is high. A major revision is needed regarding the analysis and presentation of the results, in addition to some minor revisions. I do believe that this can become a good contribution to the field.

Abstract and introduction

1) Abstract p.2: “The goal was to identify implementation barriers, strategies used to address them, and remaining challenges.” Introduction p.6: ” The aim of this study, therefore, is to gain insight into how clinicians at Vejle Hospital experience the introduction of a SDM initiative into their workflow, the training process, and the impact on their daily practice.” Discussion p.20: “The purpose of this qualitative study was to examine practitioners’ experiences of the Center’s efforts to introduce SDM and PDAs into clinical practice.” The aim/ goal/purpose of the study should be consistent throughout the manuscript.

Authors’ response:

We thank the reviewer for pointing out the inconsistencies regarding the study aim. We have adjusted the phrasing in the abstract to be consistent with the Introduction and Discussion. We have also reformulated the research question in the final paragraph of the Introduction on pg 5, lines 79-86 (see our response to comment 2 below).

2) The research question should be reported.

Authors’ response:

We have reformulated the research question in the final paragraph of the Introduction (pg 5) as follows: “The purpose of this study is to gain insight into practitioners’ views on the introduction of a SDM initiative into their workflow at Vejle Hospital, in particular: (i) their initial attitudes towards SDM as a concept; (ii) their experience of the implementation process and factors that they found helpful; and (iii) perceived outcomes.”

Methods

3) I recommend the authors to give the description of the authors’ contribution under each related section, for example “The interviews were conducted by AA..” could preferable have been described under the heading data collection. “Participants were recruited by means of purposive sampling [16] and were identified through KDS..” is related to study population and could preferable have been described under this heading. 

Authors’ response:

We have moved descriptions of each author’s contributions to the relevant subsections as suggested. We used the Vancouver recommendations as outlined by the ICMJE as a guideline (see our response to comment 4 below).

Ethical guidelines:

4) The contribution of the authors AA, KDS and HH is described but the role of the other authors is unknown. What was their contribution in this study? This is important to describe according to the Vancouver recommendations.

Authors’ response:

In terms of the Vancouver recommendations, the contribution of all authors are as follows:

Conception and design: AA, KDS, AD, LW, RF

Data acquisition: AA, KDS, KO

Data analysis: AA, HH, RF, CR

We have added a description of their involvement in the subsections ‘Study design’, ‘Data collection’, and ‘Data analysis’ respectively. Furthermore, all authors were involved in drafting/revising the manuscript and gave their approval for its submission for publication.

5) Anonymization of the participants is important in research and the authors should outline how anonymity is facilitated.

Authors’ response:

We have added a brief description of how confidentiality and anonymity were preserved in the Ethics subsection (pg 9, lines 137-140).

Data collection:

6) The interview guide is presented with two themes numbered 5.

Author’s response:

We thank the reviewer for noting the error and we have changed the final item of the interview guide to ‘(6)’ on pg 6, line 99.

7) Please report who transcribed the interviews.

Authors’ response:

We have added the information that AA transcribed the interviews to the ‘Data collection’ subsection (pg 10, line 156).

8) The authors report a study consisting of semi-structured interviews. This design is not compatible with “observations were recorded as field notes” (p.11). If the field notes are included in the data material in addition to the interviews the study design should be revised.

Authors’ response:

The field notes were recorded by AA as a reflection on the SDM process. The notes did not form a part of the data material that was analyzed under the thematic analysis method. As a result, to maintain clarity, the reference to field notes has been removed.

Analysis:

9) Please include a reference to thematic analysis method.

Authors’ response:

We have added a systematic description of the six phases of the thematic analysis method we followed, namely that described by Braun and Clarke (2006), along with a reference to the same. These details are given on pg 10-12, lines 168-199.

10) It is unclear how the field notes were analysed. Did the author mix the two different data sets, the interviews and the field notes? Can the authors elaborate this?

Authors’ response:

As mentioned in our response to comment 8, the field notes did not form a part of the data analysis and the reference has therefore been removed.

11) Please elaborate what the interpretations were based on.

Author’s response:

Analysis was conducted based on a collaborative reflexive approach based on the researchers’ interpretation of the data and prior theoretical knowledge. Our approach was inductive and prioritized engaging with the data and discuss multiple interpretations as opposed to achieving a specific consensus. This clarification has been added under the ‘Data analysis’ subsection on pg 12, lines 190-193.

12) The last section in “Data analysis” focuses on the interviews and should be moved to related section (data collection).

Authors’ response:

As suggested, we have moved the paragraph to the end of the ‘Data collection’ subsection.

Results, discussion, conclusions

13) The quote on p.16 is not identified (participant no.)

Authors’ response:

We have added the participant no. to the quote (pg 16, lines 279-280). 

14) It is unclear if the results reflect the data material (the participants’ answers of the 6 themes in the interview guide). The aim of this study was to gain insight into how clinicians at Vejle Hospital experience the introduction of a SDM initiative into their workflow, the training process, and the impact on their daily practice. Only “The challenges encountered by our participants at each phase and the strategies they used to address them are summarized in Figure 1 and elaborated below” (p.12) are presented as results. The authors are advised to look at the entire data material in light of the research question (which must be presented in the manuscript) and make an analysis were the results reflect the whole data material and the aim of the study.

Authors’ response:

Following the reformulation of the research question into three specific components in the Introduction section, we have reorganized the Results section so that the themes derived for each component are systematically presented and clarified (pg 12-22). 

15) Figure 1 presents an overview of challenges to SDM implementation and strategies to overcome them at different phases of implementation. It is unclear if these items are the themes, categories and codes. The authors are encouraged to provide a description of the coding tree. There should be clarity of themes were the themes are clearly presented in the results and clarity of categories and codes with a description of diverse cases or discussion of these in the results.

Author’s response:

As suggested, we have replaced the original Figure 1 with a thematic map showing the themes and subthemes corresponding to each of the three components of the research question (pg 13, lines 211-213). 

16) The conclusion should answer the aim and research question and must be reconsidered.

Authors’ response:

We have expanded the conclusion to include the answers to all three parts of the research question formulated in the Introduction (pg 29, lines 532-542). 

17) The patient perspective is the onus of SDM. This study reflects only the clinicians’ perspective and should have been reported as a limitation and discussed.

Authors’ response:

We thank the reviewer for the suggestion and have added the lack of patient perspective as a limitation and discuss the implications for our conclusion as well as future research (pg 28, lines 520-530).

---

## [Decision Letter · Decision Letter 1]

28 Oct 2021

Practitioners’ views on shared decision-making implementation: A qualitative study

PONE-D-21-20569R1

Dear Dr. Fijten,

We’re pleased to inform you that your manuscript has been judged scientifically suitable for publication and will be formally accepted for publication once it meets all outstanding technical requirements.

Kind regards,

Chaisiri Angkurawaranon

Academic Editor

PLOS ONE

Additional Editor Comments (optional):

Reviewers' comments:

Reviewer's Responses to Questions

**Comments to the Author**

1. If the authors have adequately addressed your comments raised in a previous round of review and you feel that this manuscript is now acceptable for publication, you may indicate that here to bypass the “Comments to the Author” section, enter your conflict of interest statement in the “Confidential to Editor” section, and submit your "Accept" recommendation.

Reviewer #1: All comments have been addressed

2. Is the manuscript technically sound, and do the data support the conclusions?

Reviewer #1: (No Response)

3. Has the statistical analysis been performed appropriately and rigorously? 

Reviewer #1: (No Response)

4. Have the authors made all data underlying the findings in their manuscript fully available?

Reviewer #1: (No Response)

5. Is the manuscript presented in an intelligible fashion and written in standard English?

Reviewer #1: (No Response)

6. Review Comments to the Author

Reviewer #1: (No Response)

7. PLOS authors have the option to publish the peer review history of their article (what does this mean?). If published, this will include your full peer review and any attached files.

Reviewer #1: No

---

## [Editor Report · Acceptance letter]

2 Nov 2021

PONE-D-21-20569R1 

Practitioners’ views on shared decision-making implementation: A qualitative study 

Dear Dr. Fijten:

I'm pleased to inform you that your manuscript has been deemed suitable for publication in PLOS ONE. Congratulations! Your manuscript is now with our production department. 

Kind regards, 

on behalf of

Dr. Chaisiri Angkurawaranon 

Academic Editor

PLOS ONE